# Effects of the Healing Beats Program among University Students after Exposure to a Source of Psychological Stress: A Randomized Control Trial

**DOI:** 10.3390/ijerph182111716

**Published:** 2021-11-08

**Authors:** Jiah Song, Wonjong Kim, Iklyul Bae

**Affiliations:** 1College of Nursing, Konyang University, 158 Gwanjeodong-ro, Seo-gu, Daejeon 35365, Korea; sjiaaah@naver.com; 2College of Nursing, Eulji University, 712 Dongil-ro, Uijeongbu-si 11759, Korea; 3College of Nursing, International University of Korea, 965 Dongbu-ro, Munsan-eup, Jinju-si 52833, Korea

**Keywords:** anxiety, autonomic nervous system, heart rate, stress, music

## Abstract

This study is a randomized pre- and post-controlled trial to determine the effects of the Healing Beats program on anxiety, autonomic nervous balance, Bispectral (BIS) index, and heart rate among university students after exposure to a source of mental stress. Data were collected from candidates who volunteered from November 2018 to May 2019 in response to recruitment announcements. The analysis was performed using data of 99 participants in three groups: 32 in an experimental group, 35 in a placebo group, and 32 in a control group. The experimental group who received treatment via the Healing Beats program exhibited a significant effect on calming anxiety, autonomic nervous balance, BIS index, and heart rate, compared with the placebo group and the control group. The group interaction also showed a significant difference. The Healing Beats program can be used as an effective intervention for sedation in clinical situations or calmness in stressful situations in everyday life. Specifically, the Healing Beats program could serve as basic data for nursing interventions, according to the stability effect in stressful situations; it can also be applied to effective nursing practice as an initial study to confirm theoretical and practical indicators.

## 1. Introduction

### 1.1. The Necessity for Research

The World Health Organization defines stress as experiencing and responding to factors that cause anxiety, dissatisfaction, agitation, tension, or obsession and as an “infectious disease of the 21st century” [1]. Furthermore, the responses to stress sources vary depending on the individual. As such, stress has recently become a critical issue in modern society [1]. According to the third stage of General Adaptation Syndrome proposed by Selye [2], even if various stress-causing factors are routinely and equally presented, those who cope with and adapt well to stress sources may not find them stressful, whereas others may find them to be severely stressful. Stress reactions have often been shown to lead to mental disorders, such as anxiety disorders and depression [3], as well as various physical reactions, such as an increase in heart rate, fever, sweating, and respiratory responses [4]. Responses to stress factors vary according to the individual’s resistance to stress, and stress resilience has become an important element for appropriate stress management [5]. Although appropriate stress may act as a dynamic force or positive influence on mental and physical health, stress sources are perceived as stress when recognized as harmful or a threat, and constant exposure threatens the individual’s well-being [6].

When stress is perceived, various negative reactions occur in the body. Psychologically, there are changes in the emotional state, including anxiety, depression, anger, and tension [7]. Physiological reactions include an increase in heart rate and muscle tension, fever, sweating, increased blood pressure, changes in the cardiovascular and digestive system, and a release of hormones of the sympathetic nervous system [8]. When acute and chronic stress is perceived, the autonomic nervous system that maintains stability as a defense against stress reacts first, promoting ACTH (adrenocorticotropic hormone) secretion and increasing the activity of the sympathetic nervous system [9]. Increased ACTH secretion leads to symptoms such as tachycardia by secreting catecholamine as an essential reaction to maintain survival when exposed to severe stress. Cortisol is also secreted, which raises the glucocorticoid concentration as a stress reaction of the hypothalamus, pituitary gland, and adrenal gland [10]. Failure to properly control these stress responses may result in severe stress and may lead to significant losses for both the individual and for society [11]. In other words, if social members’ stress is not properly managed, it can lead to drug addiction, suicidal behavior, interpersonal relationships, loss of control, and even social support [12].

Various interventions are used to relieve stress; for example, medication for sedation is most widely used. However, as the long-term use of drugs may have detrimental side effects or may cause physical damage such as resistance to drugs [13], there is a growing interest in non-pharmacological, complementary, or alternative therapies that minimize the side effects of drugs [14]. Complementary or alternative therapies emphasize integrated care for stress relief. Existing studies on complementary or alternative interventions have been conducted in the following study areas: the effects of massage on stress [15], the effects of biofeedback on stress [16], the effects of exercise on physical and mental health [17], and the effects of essential oil aroma therapy on anxiety [18]. However, these alternative therapies all have certain drawbacks. For example, stress relief through massage therapy is difficult because this intervention is provided only by a trained professional and requires special time and space. The same applies to biofeedback, because it requires special devices, facilities, and space. Using exercise as an intervention also has its difficulties, because applying this technique to individuals with physical disabilities can be challenging; it could also be greatly influenced by weather, location, and diet. Aroma therapy with essential oils is constrained as the oils are manufactured and provided by a trained professional and thus are not publicly available.

However, stress interventions using music are an ordinary and popular method that is widely accessible and non-invasive. Of late, these have been widely used as a complementary or alternative therapy that reduces anxiety, stabilizes vital signs, and relieves stress, with few side effects [19]. According to earlier studies, music affects the autonomic nervous system and has a relaxing effect on heart rate and vital signs by affecting the pituitary gland and the limbic system, which secretes endorphins [20]. Furthermore, slow music stimulates the parasympathetic nerves and stabilizes the autonomic nervous system [21]. Thus far, however, previous studies on listening to music for stress relief have been limited to the use of existing sound sources, such as classical music [22], relaxation music [23], white noise [24], or preferred music [25]. Moreover, most research and design processes to verify the effects of music listening on stress are limited to pre- and post- experimental treatment comparisons, which does not indicate the time series effect of the intervention.

Therefore, this study applied the Healing Beats program as a method of nursing intervention for relaxation. The Healing Beats program is a step-by-step method for inducing relaxation that extracts sound waves equivalent to the individual’s resting heart rate and electrocardiogram (ECG) waveform and applies them in stressful situations. This study also measured the time of stress recovery by consistently tracking the changes in time flow and presents practical measures for evidence-based integrative nursing as a clinical trial study that applies both subjective and objective tools to verify the program’s effects.

### 1.2. Purpose of Research

The purpose of this study was to identify and investigate the effects of the Healing Beats program on university students’ anxiety, autonomic nervous balance, Bispectral (BIS) index, and heart rate, with subjective and objective indicators. We performed of the Healing Beats program using a randomized control experiment with three groups of participants: a control group, placebo group, and an experimental group.

## 2. Materials and Methods

### 2.1. Research Design

This study is a randomized controlled and non-synchronized trial that compares the effects of the Healing Beats program on university students’ anxiety, autonomic nervous balance, BIS index, and heart rate after exposure to stress sources (Figure 1).

### 2.2. Selection of Study Participants

#### 2.2.1. Recruitment

Voluntary applicants were recruited through a research announcement asking for study participants at * University located in the city of *. Participants were selected based on the following inclusion criteria: (1) university students who understood and agreed on the purpose of the study, and (2) adults aged 20–60 years without hearing loss. Potential participants being treated for physical and/or mental illness and/or who were taking drugs that could affect anxiety and stress—such as anti-anxiety drugs, sleeping pills, cold medicines, painkillers, or alcohol—were excluded from the recruitment process. Written informed consent to participate in the research was provided by the university students who met the selection criteria and who agreed to participate in the study.

#### 2.2.2. Calculation of the Sample Size

The required number of participants was determined based on the F-test using G*power (V 3.1.9) software. According to the significance level (0.05), power (0.80), and effect size (0.34) of the two-tailed test calculated based on previous studies [26], the required total sample size was 87 participants. Considering a possible dropout rate of 20%, 35 participants were assigned to each group. Finally, 105 participants who met the selection criteria were recruited.

#### 2.2.3. Participant Assignment

Of the 105 participants, 35 were assigned to the experimental group, the placebo group, and the control group by double-blind block randomization using the random number generator function Excel. Double-blind block randomization is a method that prevents imbalance in the number of participants between the three groups that may occur due to simple blind randomization. The block size is determined by multiples of the number of groups. Using the Excel function mentioned above, three groups (experimental, placebo, and control) and nine methods—determined by setting the block size at six—were randomly allocated to the blocks.

While collecting data for this study, three participants were excluded from the experimental group (one refused to participate and two were on medication), and three from the control group (two refused to participate and one was on medication). Thus, the final study involved 99 participants, with 32 participants in the experimental group, 35 in the placebo group, and 32 in the control group (Figure 2).

To minimize the spread effect of treatment among participants, data were collected in the following order: first the control group; then the placebo group; and at the end, the experimental group. The pre- and post-tests were conducted by one research assistant, and a blind study concealing the experimental, placebo, and control groups from the research assistant was applied to increase the study validity.

### 2.3. Research Instruments

We used both subjective and objective measurement tools to identify the effects of the Healing Beats program on anxiety, autonomic nervous balance, BIS index, and heart rate.

#### 2.3.1. Anxiety

Anxiety was measured with numeric rating scales (NRSs) as a subjective tool. The NRS is a 10-point scale ranging from 0 = “not anxious at all” (on the left) to 10 = “extremely anxious” (on the right) that indicates the subject’s perceived level of anxiety on a horizontal line. The higher the score, the higher the subject’s level of anxiety.

#### 2.3.2. Autonomic Nervous Balance

The index value of the autonomic nervous balance was measured continuously for five minutes with the Canopy9 professional 4.0 (IEMBIO, Inc., Chuncheon, Korea), a measuring tool specifically designed for the autonomic nervous system. It was defined as the low-frequency (LF) and high-frequency (HF) values obtained from five minutes of continuous measurement. LF (0.04–0.15 Hz) represents sympathetic activity in the ANS, and it has a normal range of 5.50–7.90 ms. HF (0.15–0.4 Hz) represents parasympathetic activity, and it has a normal range of 4.19–7.23 ms. Greater values indicates greater sympathetic and parasympathetic activities, respectively. Next, the time domain and frequency domain were analyzed based on the heart rate variability. With the results, the activity value rates of the sympathetic and parasympathetic nerves were quantified with the standard leads.

The autonomic nervous balance index is higher when the sympathetic nerve activity is relatively higher than that of the parasympathetic nerve; a higher index value means exposure to mentally stressful situations [26]. In this study, the first autonomic nervous balance measurement was taken following a five-minute break after entering the laboratory. The second measurement was taken immediately after exposure to the stress source. A further eight measurements were taken after the experimental treatment at five-minute intervals for 40 min. Ten measurements were taken in total.

#### 2.3.3. BIS Index

The BIS index was measured using a BIS monitor (VISTA, Aspect Medical System, Newton, MA, USA) and sensor (Sensor, Aspect Medical System Inc., Newton, MA, USA). The index is the measurement of the double spectrophotometric coefficient produced by processing the subject’s level of relaxation and awareness into brain waves. BIS index values ranging from 90 to 100 represent the awake state, 80 to 90 represent the relaxed state, 70 to 80 represent the relaxed conscious state (deeply relaxed state), 60 to 70 represent the moderate hypnotic state (may awaken but less likely to remember), 40 to 60 represent an acute hypnotic state (the range maintained during general anesthesia), and below 40 represent a deep hypnotic state. To attach the BIS index sensor, the participant’s skin was first wiped with an alcohol swab; next, the number one electrode was attached two to three centimeters above the center of the forehead, the number four electrode on the outer edge of and parallel to the eyebrow, the number three electrode at the ipsilateral temporal area, and the number two electrode on the forehead between the number one and four electrodes. Skin resistance was verified after attaching the electrodes and measurements were taken after resistance was reduced by applying gentle pressure on the electrodes with high resistance. The first measurement was taken following a five-minute break after entering the laboratory. The second measurement was taken immediately after exposure to the stress source. An additional eight measurements were taken after the experimental treatment at five-minute intervals for 40 min. Ten measurements were taken in total.

#### 2.3.4. Heart Rate

Heart rate (beats per minute; BPM) was measured with ECG equipment connected to a Philips HP Monitor (M1106C). Heart rates ranging from 60 to 100 BPM are within the normal range, whereas a heart rate below 60 BPM indicates bradycardia, and one of more than 100 BPM indicates tachycardia and arrhythmia. The first measurement was taken following a five-minute break after entering the laboratory. The second measurement was taken immediately after exposure to the stress source. An additional eight measurements were taken after the experimental treatment at five-minute intervals for 40 min. Ten measurements were taken in total.

#### 2.3.5. General Characteristics

Survey questionnaires were used to collect the general and health-related characteristics of the participants. Six general characteristics, including age, height, weight, drinking habits, and smoking status, and three health-related characteristics, including current diagnosis, medical history, and medication status, were observed.

### 2.4. Experimental Treatment

#### 2.4.1. Laboratory Environment

The surface area of the laboratory was 19.83 m² and the indoor temperature was set at 25 °C, a suitable temperature for measuring the autonomic nervous balance, BIS index, and heart rate of human participants. The laboratory was well-ventilated and equipped with a sofa, a table, and chairs to provide a comfortable environment for the participants. It also contained a bed for experimental measurements. Monitoring equipment was installed to measure autonomic nervous system balance, BIS index, and heart rate, which were variables measured from the participants.

#### 2.4.2. Healing Beats Program

The Healing Beats program (application number 10-2018-0147801, inventor names: Ik-Ryeol Bae and Myung-Heng Heo) is a customized stress relief program that uses sound sources linked with sound waves extracted from the participant (application sound number: resting heart rate and normal ECG waveforms). This system relieves stress by extracting the participant’s heartbeat rhythm as a waveform and applying the same waveform sound source to the extracted waveform result from the healing beat. Therefore, it gradually induces relaxation of the autonomic nervous system by applying the participant’s resting BPM in stressful situations [27].

In other words, Healing Beat is an auditory solution system that relieves stress by stabilizing the balance of the autonomic nervous system. The healing beat mechanism provides a sound source configured in the same way as the heart rate and ECG waveform at rest, customized for each individual in a stressful situation. First, the program consists of the following factors: a total of 41 different BPM options ranging from 60 to 100 beats per minute, the average heart rate range of the human body; a consistent speed (one beat in four/four time is equivalent to one second); a repetitive rhythmical pattern, a major scale, a diatonic chord, and predictable ascending and descending tones. Second, the program provides the ECG frequency and amplitude closest to the resting heart rate of the subject by extracting and realizing sound waves similar to their ECG waveforms after analyzing the power changes of sinus rhythm waveforms of the normal ECG based on amplitude and frequency. The Healing Beats program is a nursing intervention method that induces gradual relaxation by applying BPM, equivalent to the subject’s average resting heart rate in stressful situations based on heart rate and ECG waveforms.

#### 2.4.3. Provision of the Stress Source

The stress stimulus was in the form of asking participants to subtract 17 cumulatively, starting from 6135 for a five-minute period using mental arithmetic [28]. The participants were instructed to start from the beginning when they made a calculation mistake and were not informed beforehand that the exercise would continue only for five minutes.

### 2.5. Data Collection

Data were collected for approximately seven months, from November 2018 to May 2019. Survey questionnaires were distributed following a five-minute break after entering the laboratory, asking the participants to report their general characteristics and prior experience of anxiety. Then, the autonomic nervous balance, BIS index in the relaxed state, and resting heart rate were measured using the Canopy9 professional, a BIS monitor, and ECG equipment, respectively. The mental arithmetic stressor was then provided to subtract 17 cumulatively from 6135 over a five-minute period, and anxiety, autonomic nervous balance, BIS index, and heart rate were measured thereafter.

After the stress source was provided, the participants in the experimental group lay on the bed with headphones (Quiet Comfort15^®^ Bose, Boston, MA, USA, 2017) on to block outside noise and listened to the Healing Beats program at 40 dB, a volume that does not represent a negative stimulus to the auditory nerves. The participants in the placebo group listened to their preferred music for 40 min using headphones at 40 dB after being exposed to the stress source. The participants in the control group were not provided with any treatment.

During the experimental treatment, post-test values were measured by automating the devices for the autonomic nervous balance, BIS index, and heart rate to take a total of eight measurements at five-minute intervals for 40 min. Self-reported NRSs were completed by the participants after the treatment to minimize exogenous variables. A small return gift was presented to the participants after data collection, and all collected data were processed for anonymity and entered into a computer program for analysis.

### 2.6. Ethical Considerations

To ensure that the ethics of research were upheld and to ethically protect the study participants, this study was submitted to and approved by the Institutional Bioethics Committee of Konyang University, where the research was conducted (approval number KYU-2018-102-01), prior to the experiment. In addition, consent was provided by the head of the organization where the experiment was conducted, after the procedure for data collection was explained. The study adhered to the recommendations of the Declaration of Helsinki and data collection was conducted in a laboratory and attended to by a nurse with more than five years of working experience and two trained research assistants. The researchers and research assistants were required to receive eight hours of training from the research manager and to be thoroughly familiar with the content. Particularly, the double-blind block randomization method was explained to the researcher in charge of collecting data to prevent information leakage. The research manager did not participate in data collection and research procedures, but conducted a preliminary survey on 12 participants (5 in the experimental group, 3 in the placebo group, and 3 in the control group) to scrutinize general issues regarding research training and data collection. The results of the inspection confirmed the proceedings of the experiment without any changes to the overall experimental treatment protocol. Moreover, although listening to music is defined as a safe method that can be used without Food and Drug Administration approval, unpleasant emotions or auditory side effects could be triggered, depending on the individual. Therefore, the experiment was to be suspended as soon as the participant reported feeling triggered and was to be transferred to a medical institution for treatment. All data collected were de-identified using anonymous number coding to prevent exposure of the participants’ personal information, secured with locks for confidentiality, and used only for research purposes.

### 2.7. Data Analysis

The collected data were analyzed using IBM SPSS Statistics 24.0 software. The general characteristics of the participants were analyzed by frequency, error, and percentage, and pre-homogeneity tests for dependent variables of the experimental, placebo, and control groups were analyzed with an X²-test and ANOVA. The effects of the experimental treatment were analyzed with ANOVA and repeated measures ANOVA. The results were analyzed with ANOVA to verify the effects on anxiety, autonomic nervous balance, BIS index, and heart rate of the three groups, and repeated measures ANOVA was used to verify the differences in response to changes of time between the three groups. When the results of the repeated measures ANOVA failed to meet the assumptions of sphericity and homoscedasticity, a Wilk’s lambda value was presented with a multivariate analysis.

## 3. Results

### 3.1. Pre-Homogeneity Test of the Experimental, Placebo, and Control Groups

#### 3.1.1. Homogeneity Test for General Characteristics

There were 99 participants in this experiment, with 32 in the experimental group, 35 in the placebo group, and 32 in the control group. The average ages in the three groups were 22.09 years, 21.21 years, and 22.13 years, respectively, whereas the average weights of the three groups were 56.13 kg, 57.34 kg, and 58.59 kg, thereby showing no significant differences. Furthermore, no significant differences were found in gender, drinking habits, and smoking status, ensuring homogeneity of the three groups (Table 1).

#### 3.1.2. Homogeneity Test for Dependent Variables

The results of the homogeneity test for dependent variables are as follows: considering anxiety, the experimental, placebo, and control groups scored 1.84, 1.86, and 1.72, respectively, and exhibited no significant differences (F = 0.10, *p* = 0.907). For autonomic nervous balance, the three groups scored 1.14, 1.12, and 1.15, respectively, and exhibited no significant differences (F = 0.08, *p* = 0.922). Regarding their BIS index, the three groups scored 99.5, 99.8, and 99.66, respectively, and exhibited no significant differences (F = 0.92, *p* = 0.403). Lastly, regarding heart rate, the three groups scored 80.88, 79.4, and 78.44, respectively, and exhibited no significant differences (F = 1.00, *p* = 0.373). These indicators confirmed the homogeneity for dependent variables of all three groups (Table 1).

### 3.2. Effects of the Healing Beats Program on Anxiety, Autonomic Nervous Balance, BIS Index, and Heart Rate

#### 3.2.1. Effects of the Healing Beats Program on Anxiety in the Experimental Group, Placebo Group, and Control Group

The measurements taken to determine the effects of the Healing Beats program on anxiety showed an increase in heart rate after exposure to the stress source (F = 0.13, *p* = 0.879) and a decrease in heart rate 40 min after termination of the experimental treatment, showing a significant difference between the experimental group (0.59), the placebo group (0.97), and the control group (1.59; F = 8.49, *p* < 0.001). The results of the repeated measures ANOVA on each group’s anxiety level did not meet the assumption of sphericity (Mauchly’s W = 0.687, *p* < 0.001); therefore, Wilk’s lambda value of multivariate tests was used and showed a significant difference in interaction between groups in response to time (F = 6.30, *p* = 0.003; Table 2).

#### 3.2.2. Effects of the Healing Beats Program on the Autonomic Balance of the Experimental Group, Placebo Group, and Control Group

The measurements taken to verify the effects of the Healing Beats program on autonomic nervous balance showed an increase after exposure to the stress source in all three groups (F = 0.05, *p* = 0.951). Significant differences between the three groups appeared 10 min (F = 9.06, *p* < 0.001) and 15 min (F = 3.97, *p* = 022) after termination of the experimental treatment. The results of ten repeated measures ANOVAs on the autonomic nervous balance did not meet the assumption of sphericity (Mauchly’s W = 0.006, *p* < 0.001); therefore, Wilk’s lambda value of multivariate tests was used. The results showed significant differences between time (F = 87.55, *p* < 0.001) and groups (F = 3.69, *p* = 0.029), but no significant difference in interaction between groups in response to time (F = 1.06, *p* = 0.390; Table 2).

#### 3.2.3. Effects of the Healing Beats Program on the BIS Index of the Experimental Group, Placebo Group, and Control Group

The measurements taken to verify the effects of the Healing Beats program on the BIS index showed an increase after exposure to the stress source in all three groups (F = 0.05, *p* = 0.951). A significant difference between the groups appeared five minutes after termination of the experimental treatment (F = 7.21, *p* = 001) and continued to appear at every time point measured: 10 min after termination (F = 13.25, *p* < 0.001), 15 min after termination (F = 26.69, *p* < 0.001), 20 min after termination (F = 46.17, *p* < 0.001), 25 min after termination (F = 33.45, *p* < 0.001), 30 min after termination (F = 37.90, *p* < 0.001), 35 min after termination (F = 35.22, *p* < 0.001), and 40 min after termination (F = 31.12, *p* < 0.001). The results of ten repeated measures ANOVAs on the BIS index did not meet the assumption of sphericity (Mauchly’s W = 0.000, *p* < 0.001); therefore, Wilk’s lambda value of multivariate tests was used, resulting in a significant difference in interaction between groups in response to time (F = 6.74, *p* < 0.001; Table 2).

#### 3.2.4. Effects of the Healing Beats Program on Heart Rate in the Experimental Group, Placebo Group, and Control Group

The measurements taken to verify the effects of the Healing Beats program on heart beat showed an increase after exposure to the stress source in all three groups. Furthermore, a significant difference in heart rate appeared 15 min after the experimental treatment between the groups (F = 5.43, *p* = 0.006). Thereafter, the heart rate of the three groups decreased, showing significant differences at the following time points: 20 min after the experimental treatment (F = 19.33, *p* < 0.001), 25 min after the experimental treatment (F = 25.64, *p* < 0.001), and 30 min after the experimental treatment (F = 11.60, *p* < 0.001). The results of ten repeated measures ANOVAs did not meet the assumption of sphericity (Mauchly’s W = 0.000, *p* < 0.001); therefore, Wilk’s lambda value of multivariate tests was used, resulting in a significant difference in interaction between groups in response to time (F = 6.57, *p* < 0.001; Table 2).

## 4. Discussion

As a result of confirming the effect of the Heating Beats program intervention on stress resilience in this study, it was found that the experimental group exhibited a significant effect on anxiety sedation, autonomic nervous system balance, BIS index, and heart rate compared to the placebo group and the control group. There were also significant differences in group interactions.

Prior to the experimental treatment, the participants were exposed to a mental arithmetic stressor, and both subjective and objective stress levels were measured after exposure to the stress source. According to existing studies, the level of stress caused by mental arithmetic stressors is equivalent to the level of stress commonly experienced in everyday life. Exposure to such a stress source increases sympathetic nerve activity [29]. The mental arithmetic stressor used in this study caused an increase in the levels of anxiety, autonomic nervous balance, and heart rate; therefore, it can be seen as an effective stress source.

To verify the effects of the program on anxiety, the participants’ anxiety levels were subjectively measured three times with NRS; before and after exposure to the stress source and 40 min after the experimental treatment. This was to minimize the interference of exogenous variables when automatically measuring the objective variables for 40 min after the experimental treatment at five-minute intervals. The anxiety level measured at the termination of the experiment was lowest in the experimental group, then higher in the placebo group, and highest control group. The results of three repeated measures ANOVAs in time series showed a significant difference in interaction between the groups in response to time. Specifically, all three groups exhibited a decrease in their anxiety levels after the experimental treatment, compared with their anxiety levels prior to treatment. This verifies the effectiveness of the Healing Beats program on reducing anxiety, compared with preferred music and no treatment; however, it should be noted that anxiety states in stressful situations may be relieved to some degree over time without intervention.

To support these results, the autonomic nervous balance was measured as an objective indicator of stress resilience. The autonomic nervous balance index is the activity value rates of the sympathetic and parasympathetic nerves quantified with standard leads based on the change in heart rate; the higher the value, the greater the exposure to mental stress situations. In this study, all three groups showed a significant decrease in autonomic nervous balance when measured 10 min and 15 min after applying the Healing Beats program. Particularly, the autonomic nervous balance value of the experimental group measured 15 min after the experimental treatment was lower than the initial value, measured at the participants’ relaxed state. This indicates the effectiveness of the program for mental stress resilience. The results measured from 20 min to 35 min after the experimental treatment showed a decrease in each group but no significant differences between the groups. The results measured 40 min after the experimental treatment showed an increase from the value measured 35 min after the treatment in all three groups. These results indicate that the time point of stress resilience can be identified by examining the effects of the program measured sequentially in time series. It also indirectly reveals that the activity values of the sympathetic and parasympathetic nerves react sensitively to environmental factors. Although the results of ten repeated measures ANOVAs on the autonomic nervous balance showed a significant difference between time and groups, there was no significant difference in interaction between groups in response to time. This indicates that the program has a limited effect on the autonomic nervous balance.

When measuring the effects of the Healing Beats program on sedation scores through brain waves with the BIS index, the experimental group showed a significant decrease compared with the placebo and control groups. In particular, the experimental group showed an acute relaxation effect, five minutes after the experimental treatment, by scoring the lowest on the BIS index (92.31), compared with the placebo group (94.03) and the control group (96.06). All three groups showed a consistent and significant decrease from the 10 to 35 minutes’ measurements after the experimental treatment, with the sedation score of the experimental group reaching a moderate hypnotic state at 30 min (69.47) and 35 min (66.84) after the treatment. Although these results are in line with results of an existing study on the effects of listening to music [30], our study confirmed the step-by-step relaxation effect of the program in stressful situations, because there was a significant difference between the experimental group and the placebo group (preferred music) and the control group (no treatment). The results of the repeated measures ANOVA used to verify the time series effect showed a significant difference in interaction between groups in response to time. Furthermore, the value measured at 40 min after the treatment was slightly higher than the value measured 35 min after the treatment in all three groups. This reaction is similar to that of the autonomic nervous balance, which shows the effects of resilience on stress relief.

Finally, according to Taelman, Vandeput, Spaepen and Van Huffel [31], heart rate increases when tension and anxiety are raised in stressful situations. Thus, the effects of the Healing Beats program on heart rate were measured as an objective assessment of stress relief. The experimental group program showed a significant decrease in heart rate from 15 min to 30 min after the experimental treatment, compared to the placebo and control groups. According to the results of tracking and monitoring the heart rate in time series, the initial heart rate of the experimental group was 80.88. After being exposed to the stress source, the heart rate increased to 113.88, and then started to show a significant decrease 15 min after applying the program, compared with the placebo and control groups. The time point at which subjects in the experimental group recovered their initial heart rate was 20 min after the experimental treatment. Although the placebo group also showed a decrease in heart rate over time, it took 40 min after the experimental treatment to reach 79.58, coming close to the initial resting heart rate (79.4). Participants in the control group did not recover a heart rate below the initial heart rate of 78, even after 40 min. The results of ten pre- and post-test repeated measures ANOVAs on heart rate showed a significant difference in interaction between the groups in response to time, confirming that the program decreases heart rate in stressful situations and has effects on stress resilience. These results are meaningful, particularly considering that the gradual stabilizing effect the program has on heart rate was verified. Moreover, the results indicated the effective time point for intervention of the program, because the autonomic nervous balance, BIS index, and heart rate of the experimental group all significantly decreased 15 min after the experimental treatment.

This study verified the effects of the program on stress resilience after exposing the participants to a mental arithmetic stressor and measured both subjective and objective variables. The subjective stress value significantly decreased after the experimental treatment, and the objective stress variables, BIS index, and heart rate significantly decreased in the experimental group when the difference in interaction between the groups in response to time was measured. The autonomic nervous balance, however, differed between the three groups in response to the time point of stress resilience, but did not differ in interaction between the groups in response to time. Rather, a partial time series effect appeared, indicating a relaxing effect.

Therefore, the Healing Beats program may be used as an effective nursing intervention to relieve stress in stressful situations. This study is particularly meaningful because it is a differentiated initial study that used the beat of music and ECG waveforms—instead of existing music therapy methods—on patients who needed relaxation. It also provides effective clinical grounds for nursing interventions by identifying the time point of stress resilience by tracking and monitoring continuous changes in time flow. Furthermore, this study is expected to lay clinical foundations for effective nursing interventions to help patients in acute stress situations.

The Healing Beats program, which is effective in reducing stress in university students, can be employed as a foundation for productive nursing interventions to help patients. However, it should be noted that this study has certain limitations. First, the experiment was conducted in a fixed laboratory setting that implies restrictions on applying everyday life factors or stressors to clinical settings. Second, this study only used NRS and did not utilize other available questionnaires to measure the subjective variables. Therefore, further studies with more participants and expanded experiments that measure additional subjective and objective variables are needed in future.

## 5. Conclusions

This study aimed to identify the effects of the Healing Beats program on university students’ everyday life and stressful situations. The results show that the program had a significant effect on calming anxiety and gradually stabilizing the autonomic nervous balance, stress variables, BIS index, and heart rate. Moreover, as a result of tracking and measuring the time series effect after the experimental treatment at five-minute intervals, the autonomic nervous balance, BIS index, and heart rate all showed a significant decrease 15 min after the experimental treatment. This verified the effective intervention time of the program to be 15 min.

The Healing Beats program may be applied not only to healthy adults, but also to students and older adults exposed to stress sources in everyday life. It can also be used in clinical settings to relieve the stress of various patients in stressful situations. Finally, the program may provide basic data for nursing interventions for relaxing effects in stress situations and can be applied in effective nursing practices as an initial study that confirms both theoretical and practical indicators.

## Figures and Tables

**Figure 1 ijerph-18-11716-f001:**
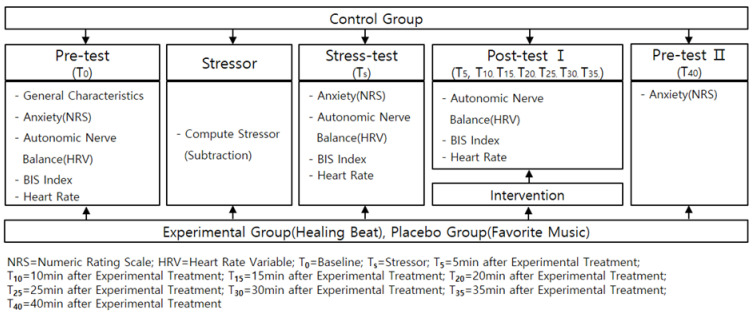
Research design.

**Figure 2 ijerph-18-11716-f002:**
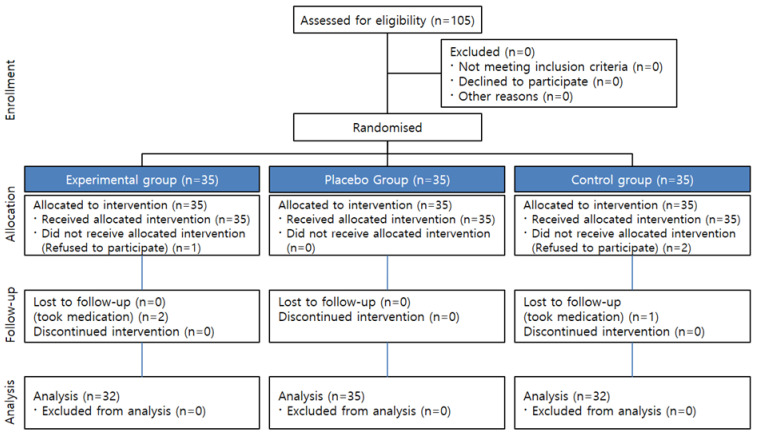
Process flow diagram.

**Table 1 ijerph-18-11716-t001:** Characteristics of the experimental, placebo, and control groups.

Characteristics/Variable	Category	Experimental Group (*n* = 32)	Placebo Group (*n* = 35)	Control Group (*n* = 32)	F/x²	*p*
		Mean ± SD/N (%)	Mean ± SD/N (%)	Mean ± SD/N (%)		
Age (years)		22.09 ± 1.20	21.51 ± 1.17	22.13 ± 2.58	1.29	0.278
Height (cm)		163.83 ± 6.19	164.14 ± 6.31	166.22 ± 7.91	1.13	0.326
Body weight (kg)		56.13 ± 10.34	57.34 ± 10.79	58.59 ± 12.80	0.378	0.686
Gender	male	6(18.8)	10(28.6)	11(34.4)	0.37	0.373
	female	26(81.3)	25(71.4)	21(65.6)		
Drinking alcohol	no	13(40.6)	12(34.3)	11(34.4)	0.83	0.835
	yes	19(59.4)	23(65.7)	21(65.6)		
Smoking	no	24(75.0)	29(82.9)	26(81.2)	0.70	0.433
	yes	8(25.0)	6(17.1)	6(18.8)		
Initial Anxiety		1.84 ± 1.65	1.86 ± 1.26	1.72 ± 1.25	0.10	0.907
Initial Autonomy nerve balance		1.14 ± 0.21	1.12 ± 0.20	1.15 ± 0.23	0.08	0.922
Initial BIS Index		99.5 ± 1.11	99.8 ± 0.91	99.66 ± 0.94	0.92	0.403
Initial Heart Rate		80.88 ± 8.44	79.4. ± 6.69	78.44 ± 5.44	1.00	0.373

SD = standard deviation.

**Table 2 ijerph-18-11716-t002:** Comparison of anxiety, autonomy nerve balance, BIS index, and heart rate of the experimental, placebo, and control groups.

Variable		Experimental Group (*n* = 32)	Placebo Group (*n* = 35)	Control Group (*n* = 32)	F	*p*	F(*p*) *
		Mean ± SD	Mean ± SD	Mean ± SD			
Anxiety	T0	1.84 ± 1.65	1.86 ± 1.26	1.72 ± 1.25	0.10	0.907	Time 33.45 (<0.001) G*T 6.30 (0.003) Group 0.60 (0.548)
	TS	4.72 ± 1.96	4.49 ± 1.69	4.59 ± 1.98	0.13	0.879	
	T40	0.59 ± 0.67	0.97 ± 0.95	1.59 ± 1.24	8.49	<0.001	
MSI	T0	1.14 ± 0.21	1.13 ± 0.20	1.15 ± 0.23	0.08	0.922	Time 87.55 (<0.001) G*T 1.07 (0.390) Group 3.69 (0.029)
	TS	2.23 ± 0.48	2.20 ± 0.46	2.23 ± 0.31	0.05	0.951	
	T5	1.76 ± 0.50	1.80 ± 0.49	2.00 ± 0.43	2.44	0.093	
	T10	1.32 ± 0.35	1.45 ± 0.47	1.75 ± 0.40	9.06	<0.001	
	T15	1.12 ± 0.19	1.14 ± 0.27	1.29 ± 0.32	3.97	0.022	
	T20	1.07 ± 0.14	1.09 ± 0.23	1.15 ± 0.29	1.10	0.339	
	T25	1.05 ± 0.15	1.07 ± 0.23	1.08 ± 0.18	0.21	0.813	
	T30	1.04 ± 0.17	1.04 ± 0.15	1.06 ± 0.17	0.23	0.799	
	T35	0.98 ± 0.13	1.00 ± 0.17	1.02 ± 0.13	0.78	0.462	
	T40	1.03 ± 0.21	1.03 ± 0.21	1.03 ± 0.11	0.01	0.993	
BIS Index	T0	99.50 ± 1.11	99.80 ± 0.63	99.66 ± 0.94	0.92	0.403	Time 142.46 (<0.001) G*T 6.74 (<0.001) Group 51.23 (<0.001)
	TS	99.81 ± 0.64	99.97 ± 0.17	99.75 ± 0.62	1.64	0.200	
	T5	92.31 ± 5.09	94.03 ± 3.59	96.06 ± 2.90	7.21	0.001	
	T10	86.22 ± 7.61	91.17 ± 5.07	93.56 ± 4.38	13.25	<0.001	
	T15	82.72 ± 6.71	88.91 ± 4.53	92.13 ± 4.18	26.69	<0.001	
	T20	77.13 ± 8.51	85.86 ± 4.99	91.66 ± 3.90	46.17	<0.001	
	T25	71.97 ± 10.19	80.08 ± 7.54	88.09 ± 5.15	33.45	<0.001	
	T30	69.47 ± 9.45	76.69 ± 7.45	86.47 ± 6.34	37.90	<0.001	
	T35	66.84 ± 9.62	74.89 ± 6.89	83.44 ± 7.00	35.22	<0.001	
	T40	71.56 ± 8.04	76.46 ± 8.59	86.56 ± 6.42	31.12	<0.001	
HR	T0	80.88 ± 8.44	79.40 ± 6.69	78.44 ± 5.44	1.00	0.373	Time 71.81 (<0.001) G*T 6.57 (<0.001) Group 3.62 (0.030)
	TS	113.88 ± 19.39	115.89 ± 15.50	109.88 ± 15.37	1.10	0.338	
	T5	103.22 ± 14.16	109.60 ± 12.89	106.88 ± 12.08	2.00	0.141	
	T10	90.69 ± 8.67	92.97 ± 7.46	95.06 ± 5.58	2.84	0.064	
	T15	86.03 ± 8.87	89.31 ± 7.17	92.22 ± 6.31	5.43	0.006	
	T20	8.00 ± 7.99	85.20 ± 5.25	89.94 ± 5.71	19.33	<0.001	
	T25	77.19 ± 8.11	82.43 ± 4.43	87.81 ± 4.67	25.64	<0.001	
	T30	76.78 ± 8.11	80.77 ± 5.55	84.47 ± 5.16	11.60	<0.001	
	T35	78.19 ± 7.74	80.38 ± 6.44	81.44 ± 5.11	2.07	0.132	
	T40	78.66 ± 7.44	79.58 ± 6.28	79.44 ± 5.18	0.20	0.821	

SD, standard deviation; *, repeated measures ANOVA; G*T, group * time; T0, baseline; TS, stressor; T5, 5 min after experimental treatment; T10, 10 min after experimental treatment; T15, 15 min after experimental treatment; T20, 20 min after experimental treatment; T25, 25 min after experimental treatment; T30, 30 min after experimental treatment; T35, 35 min after experimental treatment; T40, 40 min after experimental treatment.

## Data Availability

No new data were created or analyzed in this study. Data sharing is not applicable to this study.

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
