# Peer review of "Effects of the Healing Beats Program among University Students after Exposure to a Source of Psychological Stress: A Randomized Control Trial"

_ijerph, 2021, doi:10.3390/ijerph182111716_

Round 1
Reviewer 1 Report
General comments
Thank you for allowing me to review this paper. This study is novel and interesting, I would have liked to hear the authors’ opinions on how it could be used in clinical populations more. There are some changes in the paper that I think would make it stronger, see below.
Specific comments
=============
Major comments
---------------------
- You talk about alternative therapies for stress reduction – what about psychological therapies such as CBT? And further, why do alternative therapies need to be used in conjunction with medication and psychological therapies?
- In the introduction you mention that the program extracts sound waves equivalent to the participant’s resting heart rate – what is the rationale for this? You explain it in the method but it needs to be clear sooner
- What did the placebo group receive?
- The discussion should be organised in a way where the important conclusions are discussed first, at the moment it is hard for readers to take away the important findings
Minor comments
---------------------
- You need to define ACTH before using the anacronym
- You state “Failure to properly control these stress responses may result in severe stress and may lead to significant losses for both the individual and for society” in the introduction – this is vague, can you be more specific as to the losses?
- I’m interested – are there any side effects for music therapy?
- You need to outline what BIS and autonomic balance is in the introduction for readers who are not across this term
- Rather than saying “without disease that may affect one’s hearing” in the method – why not instead say that they didn’t have hearing loss?
- Does caffeine intake affect heart rate? Did you ask participants this?
- The authors go over the method again in the discussion and I don’t think this is necessary
Reviewer 2 Report
This paper is well written. Only few suggestions are listed as the following:
1.Research hypothesis looks redundant.
2.the reliability and validity of instruments should be described.
3. 2.4.3. Provision of the stress source---please offer the validity or evidence to support the relevance or validity of your assessment by asking participants to subtract 17 cumulatively, starting from 6,135 for a five-minute period using mental arithmetic.
4. Personally, the Healing Beats program is not clear or convinced to me for its mechanism in relieving stress by using sound sources linked with sound waves extracted from the participant. What's difference between this study and Ref 26? Both are RCT testing the HBP and look alike?
Reviewer 3 Report
Work very interesting, innovative, original. The study should be further conducted e.g. in other research/scientific centers. The subject matter is important from the perspective of public health and health psychology.
The scholarly approach to the topic, the fulfillment of all criteria required for the research process, and the overall appearance of the paper allows me to accept the paper for publication after addressing the comments/observations below.
It is worth referring to my comments:
- I suggest changing the title, e.g., Effects of the Healing Beats program among University Students after exposure to a source of psychological stress.
- line 52, 52 - resolve acronym "ACTH"
- Table 2 - unnecessary
